



# 1 40 years High Arctic climatological dataset of the Polish Polar Station

# 2 Hornsund (SW Spitsbergen, Svalbard)

Tomasz Wawrzyniak, Marzena Osuch
Institute of Geophysics, Polish Academy of Sciences, Warsaw, Poland
*Correspondence to*: Tomasz Wawrzyniak (tomasz@igf.edu.pl)
**Abstract.** The article presents the climatological dataset from the Polish Polar Station Hornsund located in the SW part of
Spitsbergen - the biggest island of the Svalbard Archipelago. Due to a general lack of long-term in situ measurements and
observations, the high Arctic remains one of the largest climate-data deficient regions on the Earth, so described series is of
unique value. To draw conclusions on the climatic changes in the Arctic, it is necessary to analyse the long-term series of
continuous, systematic, in situ observations from different locations and comparing the corresponding data, rather than rely on
the climatic simulations only. In recent decades, rapid environmental changes occurring in the Atlantic sector of the Arctic are
reflected in the data series collected by the operational monitoring conducted at the Hornsund Station. We demonstrate the
results of the 40 years-long series of observations. Climatological mean values or totals are given, and we also examined the
variability of meteorological variables at monthly and annual scale using the modified Mann-Kendall test for trend and Sen's
method. The relevant daily, monthly, and annual data are provided on the PANGAEA repository
(https://doi.pangaea.de/10.1594/PANGAEA.909042, Wawrzyniak and Osuch, 2019).

## 18 1 Introduction

If the Arctic climate changes are considered, then the long term operational monitoring of meteorological variables including
reliable observations and measurements is obligatory. Weather conditions are crucial drivers that have feedback on many
environmental components, and it is crucial to have a relevant dataset of atmospheric observation data if analysing the
variability and fluctuations of climate at any given location. Climate change in the Arctic reflects a global warming trend, but
there are regional differences throughout the area. The climatic characteristics are primarily determined by astronomical
factors, but there are differences in the mechanisms that cause a regional warming trend and determine their magnitude. The
presence of solar radiation, modified by the degree of cloudiness and type of clouds is the main factor influencing the transfer
of energy. During the polar night, the sole source of energy is the dynamic advection of heat from the ocean and atmospheric
circulations. The growing number of positive annual air temperature anomalies in the Arctic varies substantially within the




region, with the strongest changes observed in the Atlantic sector (Przybylak, 2016). Here, the Greenland Sea to the West of
Svalbard is dominated by the West Spitsbergen Current, carrying warm (3–6°C) and salty (>35‰) Atlantic waters towards the
Fram Strait. In this region, this flow is over 200 km wide and strongly influences the air temperature in the SW Spitsbergen
area, especially during the winter (Walczowski et al., 2017). The specific maritime and mild climatic conditions are also
influenced by local and regional factors as the presence of glaciers, orography of the terrain, and location near the seashore.
The climatic variables such as air temperature, humidity, and precipitation vary significantly across the archipelago (Nordli et
al., 2014; Osuch and Wawrzyniak, 2017a) as well as around the Hornsund Fjord (Araźny et al., 2018). Long-term, high-quality,
in-situ consistent meteorological observations have been collected at the Hornsund Station located at the northern shore of this
fjord. Relatively to the other parts of the Arctic, air temperatures in Hornsund are the highest at this latitude and their observed
changes are the largest on Earth. Recently observed along the western Spitsbergen: higher air temperatures and higher liquid
precipitation have many environmental implications, leading to prolongation of the ablation season (Osuch and Wawrzyniak,
2017b), negative mass balance of glaciers (Van Pelt et al., 2019), and permafrost degradation (Wawrzyniak et al., 2016).
**2 Study area**

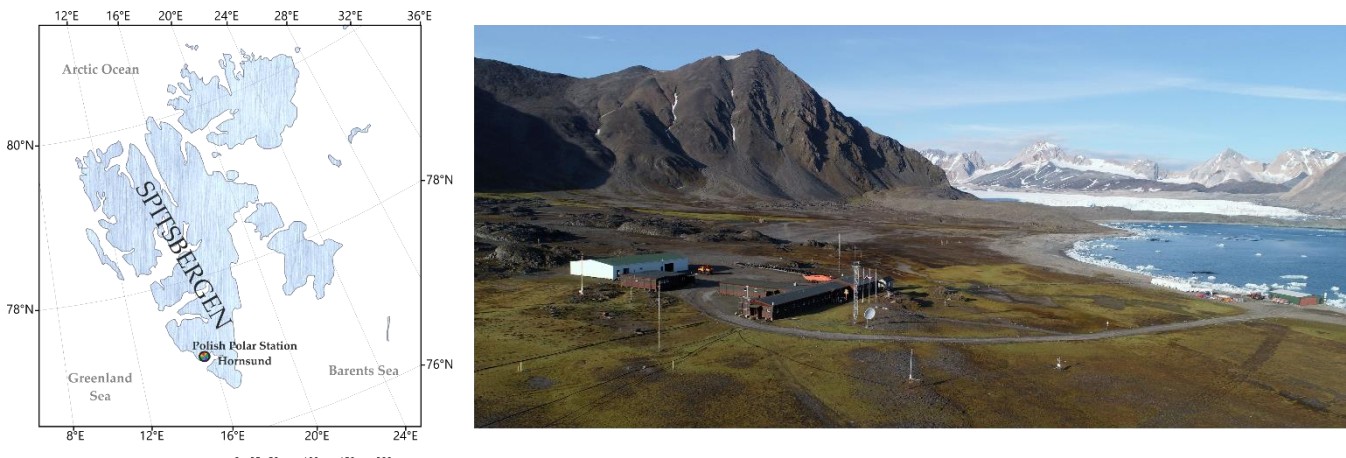


**Figure 1 Polish Polar Station Hornsund on Spitsbergen**
The Stanisław Siedlecki Polish Polar Station in Hornsund (77°00'N 15°33'E) located 300 m from the shore of Isbjørnhamna
Bay of the Hornsund Fjord in SW Spitsbergen (Fig. 1), was established during the International Geophysical Year in 1957.
Since 1978 it conducts year-round scientific research and is the northernmost permanent Polish scientific site, that throughout
the years became a modern interdisciplinary scientific platform that carries out research projects aimed at a better
understanding of the functioning of the arctic nature and the changes it undergoes. The Hornsund Fjord is approximately 35
km long and approximately 14.5 km wide at its mouth to the Greenland Sea. The coastline of Hornsund is diversified, with
multiple bays and glaciated valleys. A recent expansion of the ice-free areas is observed in Svalbard, with the most significant



retreats of the tidewater glaciers (Błaszczyk et al., 2013), so the recognition of the changes in the functioning of the
environmental system becomes more and more essential. The Station is set on a marine terrace at 10 m a.s.l., covered with sea
gravel, raised during Holocene (Lindner et al., 1991), covered by a diversity of tundra vegetation types. The slopes of the
nearest mountain ranges Fugleberget (569 m) and Ariekammen (517 m) are located 1 km north from the Station. Around 800
m NE from the Station lies the lateral moraine of Hansbreen glacier, which was at its maximum Holocene extent in Little Ice
Age (Błaszczyk et al., 2009). Recently the distance from the Station to the front of Hansbreen is around 2.5 km. The ground
here has a continuous permafrost layer down to more than 100 m deep (Wawrzyniak et al., 2016).
At Hornsund meteorological site indexed by international numbering system 01003 (www.ogimet.com), managed by the
Institute of Geophysics Polish Academy of Sciences, since July 1978 year-round, systematic, continuous measurements and
observations at WMO standards have been conducted. The results of automatic measurements and visual observations are sent
as the SYNOP-code to WMO database every 60 minutes and 3 hours, respectively. Since January 2001 most of the traditional
instruments were replaced by an automatic weather station with Vaisala QLC-50 logger. The sensors of the new system have
been installed on meteorological mast, situated 160 m SW of the main Station building. In July 2009 Norwegian
Meteorological Institute installed an automatic Vaisala weather station that is operating simultaneously at the same mast. In
September 2016 Vaisala QLC-50 was replaced by Vaisala MAWS 301. A comprehensive description of measurements and
instruments can be found in collective work edited by Marsz and Styszyńska (2013) and in **Table 1**. Although the time series
of the data stretches up to July 1978, here we analyse the variability of climatic conditions over the period 1979–2018 and in
some cases 1983-2018, based on the availability of observations without gaps. The daily, monthly, and annual averages or
sums and the extreme range (min and max), computed from observations are provided in PANGAEA repository (Wawrzyniak
and Osuch, 2019).

## 3 Meteorological variables

Inter-seasonal weather fluctuations are determined by the changing Arctic climate system and atmospheric circulation. The
changing global climate also modifies regional conditions. Weather conditions are crucial factors that have local feedback on
many environmental components. Meteorological variables collected at the Hornsund Station help to characterise the climate
variability in this part of the Arctic and for a long time have been the background for multiple studies conducted in the SW
Spitsbergen (Osuch and Wawrzyniak, 2017b, Wawrzyniak et al. 2017). Due to the diurnal variability of all meteorological
variables, in this study, we use descriptive statistic methods to present the course and variation of multiple parameters. For
most meteorological parameters, monthly mean values are calculated from daily mean values which are retrieved using the 3-
hourly values (eight values a day, between 00:00 and 21:00 UTC); in case of precipitation 6-hourly values (12:00, 18:00, and
00:00, 06:00 UTC of the following day); and daily sum of total solar radiation from Campbell–Stokes recorder obtained at the
midnight.

**Table 1 Meteorological data measured at Hornsund including variables, current sensors, the period of operation, height, units, and**
**their annual averages or sums.**

| Variable | Location | Sensor | Period of operation | Height | Unit | Mean/sum |
|---|---|---|---|---|---|---|
| Air temperature (TA) | 77°00'1.261" N 15°32'12.267" E | A traditional thermometer in Stevensons screen, Vaisala HMP 45D, HMP155 | 1979-2018 | 2 m agl | [°C] | TAmax=-1.3 TAmean=-3.7 TAmin=-6.0 |
| Relative humidity (RH) | 77°00'1.261" N 15°32'12.267" E | Hygrometer HMP45D | 1979-2018 with gap 01.07.1982-16.08.1982 | 2 m agl | [%] | 79.7% |
| Precipitation | 77°00'5.734" N 15°32'17.077" E | Hellmann Rain Gauge | 1979-2018 | 1 m agl | [mm] | 478 mm |
| Atmospheric pressure (PA) | 77°00'1.261" N 15°32'12.267" E | PTB200A | 1983-2018 | Reduced to the sea level | [hPa] | 1008.7 hPa |
| Wind speed (WS) and direction (WD) | 77°00'1.261" N 15°32'12.267" E | Fuess 90z wind meter Vaisala WAA151 Vaisala WMT702 | 1983-2000 2001-2016 2017-2018 | 10 m agl | [m/s] | 5.5 |
| Sunshine duration (SD) | 77°00'5.935" N 15°32'14.3" E | Campbell–Stokes Heliograph | 1979-2018 | 2 m agl | [h] | 1030.8 |
| Cloudiness | On location | Visual observations | 1983-2018 | | [octas] | 5.85 |





| | | | | | | |
|---|---|---|---|---|---|---|
| Visibility | On location | Visual observations | 1983-2018 | | [marine scale] | 7.40 |

## 3.1 Air temperature


Air temperature (TA) can be presumed to be one of the most sensitive indicators of climatic changes. The time series of daily
TA from the Hornsund Station covers the period 1979 to 2018. In the case of daily mean TA, there are no gaps in data while
for maximum and minimum daily TA data for 01.09.1979, 29.02.1980, 15.06.2012, and 19.06.2017 are missing. Figure 2a
presents the variability of the annual mean of minimum, mean, and maximum TA in 1979-2018 at the Hornsund Station. An
upward trend is clearly visible for the three analysed variables. The significance of the trend was estimated by the modified
Mann-Kendall test (Mann, 1945; Kendall, 1975; Hamed and Rao, 1998) taking into account autocorrelation of time series. The
slope of the trend was estimated using Sen's method (Sen, 1968), where the slope is calculated as a median of the slopes of all
pairs of points. The outcomes of the modified Mann-Kendall indicated that the trends are statistically significant; the estimated
p-value is very small (less than 1e-07) for three presented variables. The estimated slope of trend equal to 1.34, 1.14, and
$1.00°C$/decade for minimum, mean, and maximum TA respectively. These are the highest increases of TA on the planet, and
the rest of the world is not expected to experience such changes until the end of this century (Hanssen-Bauer et al. 2019).
The results of trend analyses for mean monthly TA (min, mean, and max) are presented in **Table 2**. In almost all months there
are statistically significant trends except March. In all analysed cases the estimated slope of trend has positive values and
indicated the increase in TA. A comparison of the results between variables shows that the largest changes were found for
minimum daily TA (1.34°C/decade) while the lowest for the maximum daily TA (1.0°C/decade). Taking into account changes
between months, the largest changes were estimated for January, February, and December (larger than 2.0°C/decade for
minimum and mean daily TA). The smallest statistically significant are trends in July and August with slopes of the trend
around 0.3°C/decade.
Figure 2b shows the boxplots of monthly averages of minimum, mean and maximum daily TA from the period 1979-2018.
The variability of TA depends on the season, with the highest amplitudes during winter months. Summer TA is rather constant,
with monthly means reaching usually slightly below 5.0°C. Average monthly TA during winter and early spring usually drop
below -10.0°C. The results are in general accordance with observations made at other arctic stations and reveal that winter is
characterised by the highest variability of TA. The amplitude between the extreme high and low in this season may be several
times higher than in summer. These fluctuations are determined by the relatively stable anticyclonic subsidence with extreme
cold and the turbulent cyclonic disturbances that bring higher temperatures, greater cloudiness, and heavy precipitation. The
lowest recorded TA measured at a 2 m height above solid ground at Hornsund Station was -35.9°C on 16.01.1981, while the
absolute maximum 15.6°C on 31.07.2015. Mean annual air temperature (MAAT) in long-term 1979-2018 is -3.7°C. The
average coldest month is March with mean TA -10.2°C, and on the average warmest month is July with the mean TA of 4.6°C.



The coldest month on record with mean -17.9°C was January 1981, and the warmest June 2016 with mean 6.3°C. Additionally,
in the data set we also provided monthly and annual positive (PDD) and negative degree days (NDD), calculated as the sum
total of daily mean temperatures above or below the 0°C respectively.

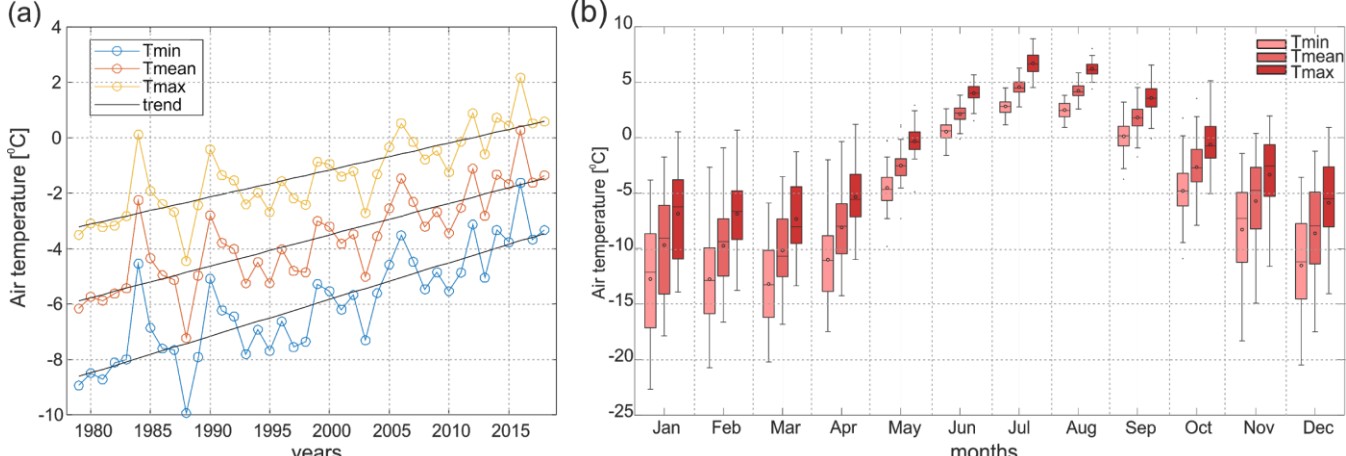


**Figure 2 (a) Variability of an *annual* mean of min, mean, and max air temperatures in 1979-2018. (b) Variability of the *monthly***
**mean of min, mean, and max air temperatures in 1979-2018. On each box, the central line indicates the median, the circle represents**
**the mean, and the bottom and top edges of the box indicate the 25th and 75th percentiles, respectively. The whiskers extend to the**
**most extreme data points not considered as outliers, and the outliers are plotted individually as dots.**
**Table 2 The slope of the trend in monthly and annual data estimated by Sen's method in the period 1979-2018 for air temperature**
**and sunshine duration and in 1983-2018 for other variables. The results of trend analysis by modified Mann-Kendall method to**
**account for autocorrelation in the time series. \* denotes lack of statistically significant trend at the 0.05 level.**

| month | TAmin [°C/dec] | TAmean [°C/dec] | TAmax [°C/dec] | RH [%/dec] | Precip [mm/dec] | PA [hPa/dec] | WS [m//decs] | SD [h/dec] | Cloudiness [octas/dec] | VV [-/dec] |
|---|---|---|---|---|---|---|---|---|---|---|
| JAN | 2.70 | 2.29 | 1.86 | -0.33* | 3.51 | 1.54 | -0.14* | 0.00* | 0.12* | -0.03* |
| FEB | 2.46 | 2.14 | 1.74 | -0.41* | 1.48* | 0.50 | 0.06* | 1.38 | -0.01* | -0.06* |
| MAR | 0.74* | 0.57* | 0.34 | -1.13 | -1.61* | -0.27 | -0.13* | 5.52* | 0.04* | 0.15 |
| APR | 1.37 | 1.00 | 0.79 | -0.64* | -0.41* | -0.66 | -0.12* | 2.36* | 0.16* | 0.11* |
| MAY | 0.97 | 0.70 | 0.49 | -0.27* | 1.92* | -0.10 | 0.40 | -10.07* | 0.00* | -0.06* |
| JUN | 0.61 | 0.52 | 0.54 | -0.65 | -3.23* | 0.27 | 0.23* | -1.80* | 0.11* | 0.13* |



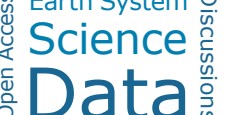

| | | | | | | | | | |
|---|---|---|---|---|---|---|---|---|---|
| JUL | 0.36 | 0.27 | 0.31 | 0.26* | 1.11* | 0.67 | 0.12* | -5.92* | 0.12* | -0.07* |
| AUG | 0.37 | 0.33 | 0.33 | 0.00* | 7.49* | 0.30 | 0.09* | 5.18* | 0.02* | -0.02* |
| SEP | 0.73 | 0.67 | 0.62 | 0.62* | 19.67 | 0.18 | 0.50 | -1.33* | 0.16* | -0.10* |
| OCT | 1.16 | 1.07 | 1.01 | 1.50 | 13.53 | 1.15 | 0.05* | 0.79* | 0.28* | -0.08* |
| NOV | 1.73 | 1.46 | 1.37 | 0.30* | 5.43* | -0.60 | 0.24* | 0.00* | 0.26 | -0.17 |
| DEC | 2.56 | 2.39 | 2.18 | 0.18* | 5.24* | -0.95 | -0.06* | 0.00* | 0.13 | -0.10* |
| annual | 1.34 | 1.14 | 1.00 | 0.10* | 61.60 | 0.25 | 0.08* | -8.39* | 0.12 | -0.02* |

## 3.2 Air humidity

The water vapour drives multiple atmospheric processes and has a significant influence on the global climate. It is the main greenhouse gas, affecting surface by feedback cycle through changing energy balance through radiative fluxes and cloud formation. According to general concepts, the Arctic warming of recent decades is accompanied by the hydrological cycle intensification (Vihma et al., 2015). To understand the variability of water vapour concentration and its causes is highly important, especially for climate studies as well as in water balance calculations. At the Hornsund Station, the air humidity is measured recently by sensor HMP155 that replaced the previously used HMP45D sensor. The observations cover period 1979-2018, but measurements were performed four times a day (0, 6, 12, 18 UTC) within the periods: 1.07.1978-26.07.1981 and 16.08.1982-31.07.1986, two times a day (6 and 18 UTC) from 27.07.1981 to 30.06.1982, eight times a day (0, 3, 6, 9, 12, 15, 18, 21 UTC) since 1.08.1986. Daily time series of the relative humidity (RH) was calculated as a mean of all available measurements within particular day. There is a gap in the measurements from 01.07.1982 to 15.08.1982. Therefore the trend analyses were performed for the period 1983-2018.

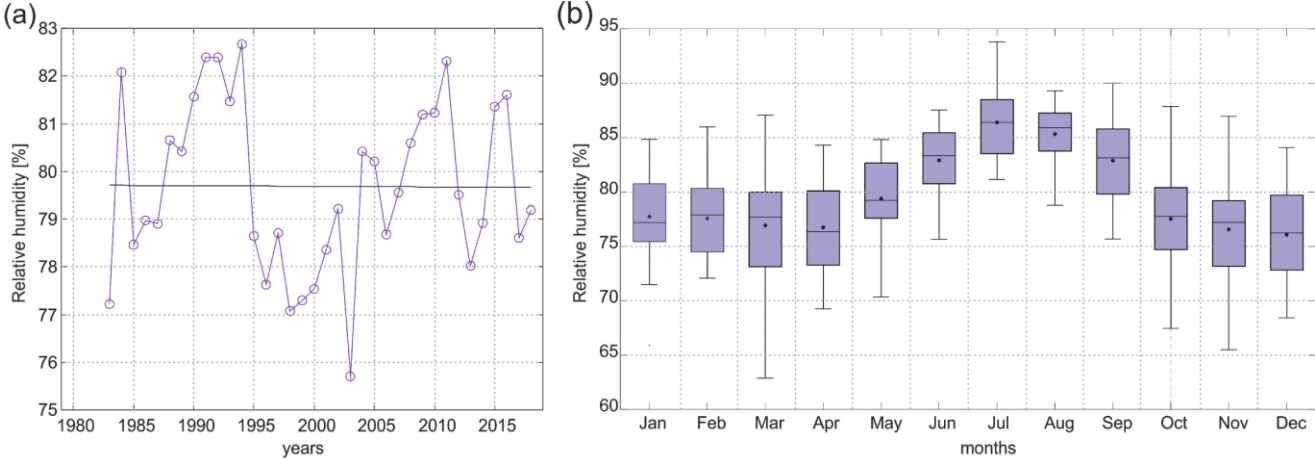

**Figure 3 (a) Variability of an annual mean of relative humidity in 1983-2018 at Hornsund. (b) Variability of mean monthly relative humidity in 1983-2018 at Hornsund.**

The variability of the annual mean RH in the period 1983-2019 is presented in Figure 3a. The average over the period 1983-2018 is 79.7%. The range of variability is from 75.7% (2003) to 82.7% in 1994. The trend analyses indicated a lack of a statistically significant trend.

The course of the monthly mean RH in the period 1983-2018 is presented in Figure 3b. Higher values of mean RH are observed in warmer months of the year and lower during winter. Such high values are attributed to continual dominance of marine air masses. The annual course of the RH is strongly connected with the air temperature and shows typical variability. It generally increases with warmer air temperatures. However, most of the trends are not statistically significant at the 0.05 level except March, June, and October.

The analyses at daily time scale indicated that drops of RH below 50% are recorded rather sporadically, although these can occur throughout the year. Such situations are connected with advection of strongly cooled air masses, foehn effects, or katabatic winds from Hansbreen (Marsz and Styszyńska, 2013). The minimum of observation reached 24% on 15.01.1981. The maximum of the observed RH is equal to 100%. Such conditions occurred 27 times in the period 1979-2018.

**3.3 Precipitation**

In the case of precipitation, the daily sum at the Hornsund Station is calculated from four measurements obtained from Hellmann Rain Gauge at 12:00, 18:00, and 00:00, 06:00 of the following day, with the orifice 200 cm², placed 1 m above the ground level. The time series of the daily sum of precipitation cover period 1979-2018 with the gap in July 1982.

The influence of the West Spitsbergen Current creates in SW Spitsbergen region a relatively moist climate which is clearly reflected in the amount of precipitation. In comparison to the other meteorological stations in Spitsbergen (Osuch and Wawrzyniak, 2017a; Hanssen-Bauer et al. 2019), the annual amount reaching 477 mm is the highest. The variability of the annual sums of precipitation in the period 1983-2018 is shown in Figure 4a. The amount of precipitation varies from 230 mm



in 1987 to 805.5 mm in 2016. The trend analyses indicated large changes, an increase of 61.6 mm/decade for the annual sums
of precipitation.

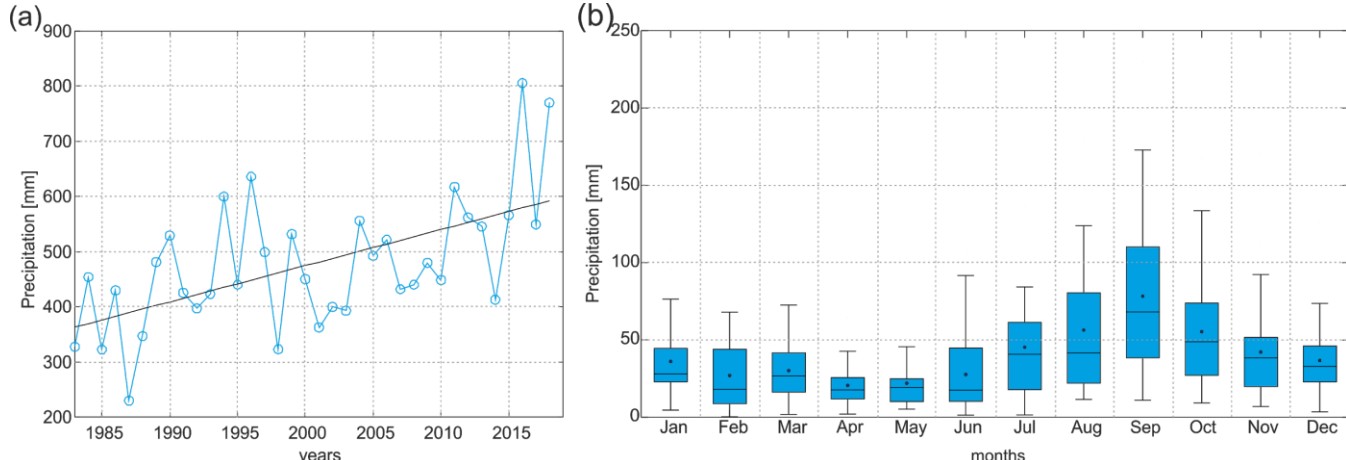


**Figure 4 (a) Variability of annual sums of precipitation in 1983-2018 at Hornsund. (b) Variability of mean monthly sums of**
**precipitation in 1983-2018 at Hornsund.**
The annual course of monthly sums of precipitation from the period 1983-2018 is presented in Figure 4b. The driest months
are April and May with the average 23 and 24 mm respectively. The highest precipitation is recorded in September reaching
on average 75 mm. Trend analyses presented in **Table 2** indicated statistically significant changes in January (3.51
mm/decade), September (19.67 mm/decade), and October (13.53 mm/decade).
**3.4 The atmospheric pressure**
The measurements of the atmospheric pressure (PA) at Hornsund started in July 1978. In the beginning, PA was measured
with a mercury barometer every 3 hours. Since 2001 measurements have been conducted every 60 seconds with a Vaisala
PTB200A sensor. The lowest recorded PA reduced to sea level at Hornsund Station was 982.2 hPa on 30.08.1994, while the
absolute maximum 1028.5 hPa on 07.08.1987. Mean annual PA in long-term 1983-2018 is 1008.7 hPa and its variability is
presented in Figure 5a. It is visible an increasing trend (0.25 hPa /decade) but not statistically significant (pval>0.05).



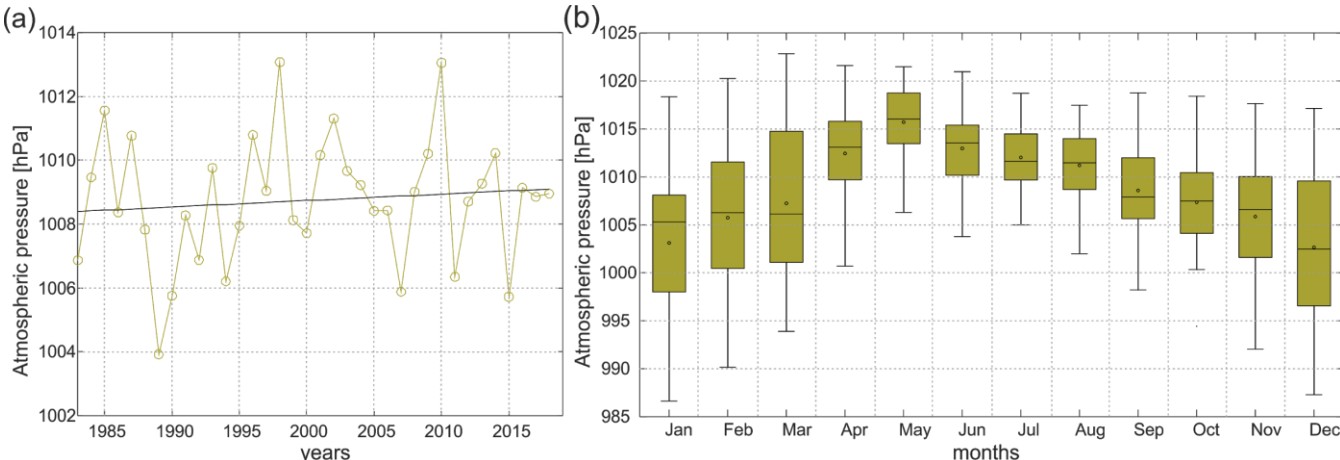

**Figure 5 (a) Variability of mean annual air pressure reduced to sea level in 1983-2018 at Hornsund. (b) Variability of mean annual air pressure reduced to sea level in 1983-2018 at Hornsund.**

Figure 5b shows the variability of the mean monthly PA over period 1983-2018. Well pronounced seasonality is visible, with mean monthly pressure higher than 1010 hPa from April to August. The month with the lowest mean PA is December with mean 1002.7 hPa, and the month with the largest PA is May with the mean of 1015.7 hPa. The variability of mean monthly PA within the observation period also is visible with the largest variability in January and February (larger than 30 hPa) and the smallest in July (13.7 hPa). The trend analyses of mean monthly PA resulted in a lack of statistically significant trend for all months.

### 3.5 Wind speed and direction

The wind is a result of atmospheric circulation and is highly correlated with the intensity of cyclonic activity (Przybylak 2016). The wind regime results from the latitudinal shape of the Hornsund fjord, location near the seashore and local topography. The measurements of wind speed (WS) and wind direction (WD) were performed at Hornsund with different sensors: 1978-2000 with the Fuess 90z wind meter, 2001-2016 with Vaisala WAA151 for direction and wind speed, since 2017 with Ultrasonic Wind Sensor WMT702. At Hornsund Station the height of the anemometer is 10 m above the ground, around 20 m above sea level. WS is measured with an accuracy of 0.1 m/s and WD with 5°. The wind rose for the Hornsund station is presented in Figure 6. Winds blowing from the East, along the fjord, are prevailing.



192

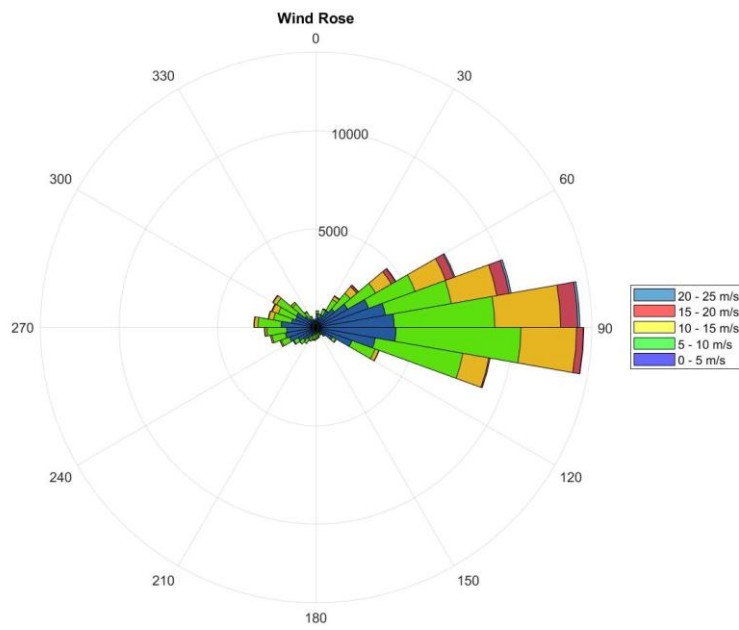

193

**Figure 6 The wind rose for the Hornsund station for the period 1983-2018.**

The variability of the mean annual WS at Hornsund in the period 1983-2018 is shown in Figure 7a. The average over the period 1983-2018 is equal to 5.5 m/s. The lowest values of WS was observed in 1985 (4.8 m/s) while the largest in 1998 (6.3 m/s). There is a lack of a statistically significant trend in mean annual WS.

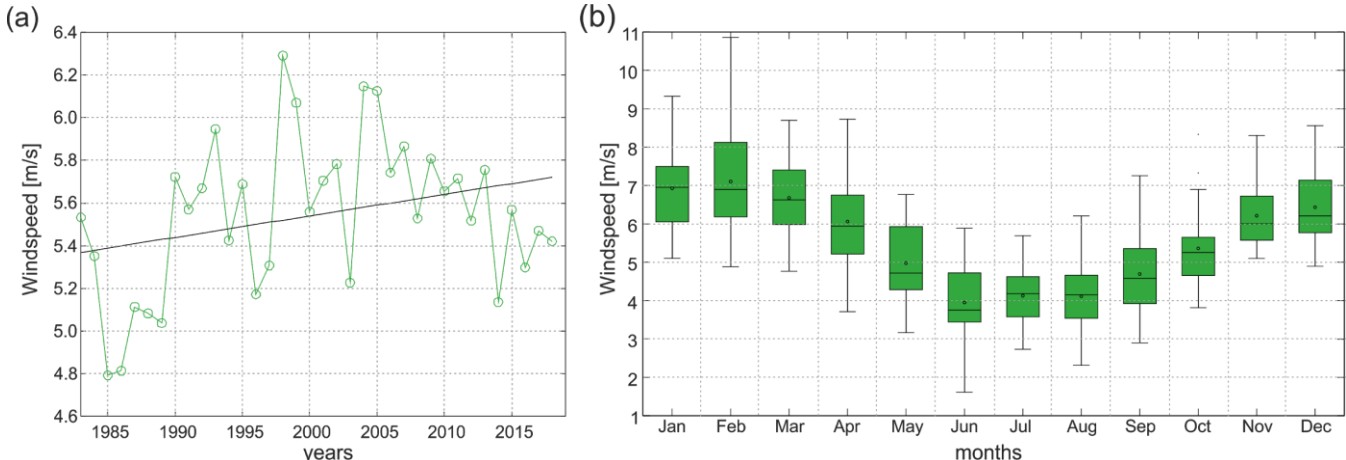


**Figure 7 (a) Variability of mean annual wind speed in 1983-2018 at Hornsund. (b) Variability of the mean monthly wind speed at Hornsund in the period 1983-2018.**



The variability of mean monthly WS in the period 1983-2018 is presented in Figure 7b. WS regime is well visible with smaller
average values during summer months (minimum 4.0 m/s in June) and larger average values during winter (maximum 7.1 m/s
in February). Such variability is a result of the extreme cyclone events that often occur during arctic winters (Rinke et al. 2017).

**3.6 Sunshine duration**

Sunshine duration (SD) is an important meteorological variable that allows the analysis of the atmospheric energy balance
widely used in climate research. Daily SD is measured at Hornsund using a Campbell-Stokes sunshine recorder (CS). It uses
a direct optical method with the heat energy of the Sun's direct radiation burning the card. Such traditional sunshine recorder
has been in service worldwide since the nineteenth century and although there are multiple automatic radiometers used
simultaneously at the Hornsund Station, the longest data is recorded by CS. The time series of sunshine duration cover period
1983-2018. At the Hornsund Station, the polar night lasts 104 days (October 31 – February 11), while the polar day lasts 117
days (April 24 – August 18).
Figure 8a shows the variability of the annual sums of SD at Hornsund in the period 1979-2018. The mean value is 1030.8 h
that is about 28% of the potential SD calculated for the Station (Wojkowski et al. 2015). The large span in the annual SD is
visible. The minimum value (755.4 h) was observed in 1994 and maximum (1325.6 h) in 1985. The slightly decreasing trend
in SD is visible but not statistically significant at the 0.05 level.

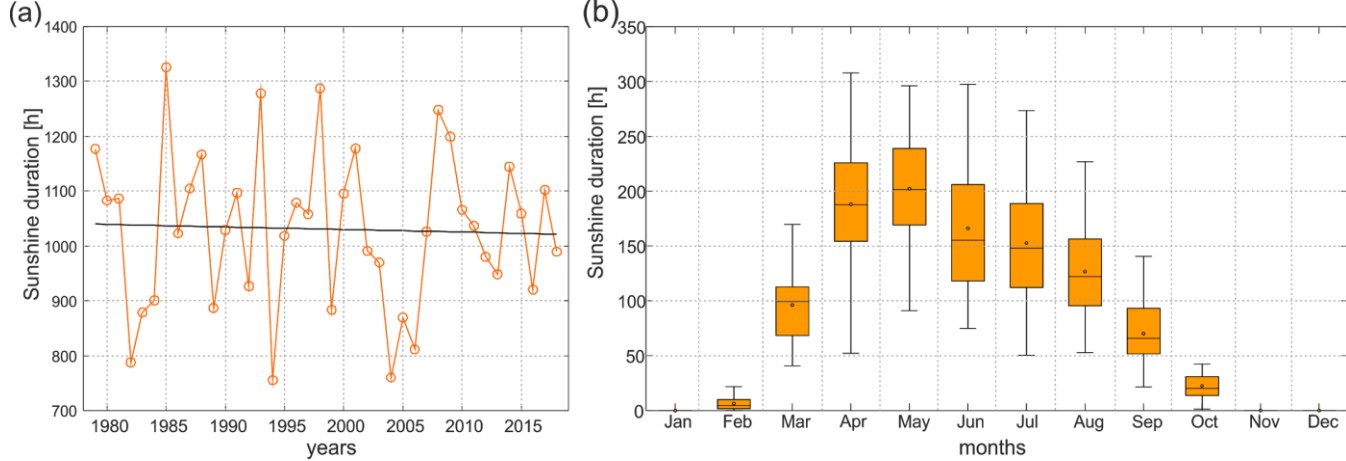

**Figure 8 (a) Variability of mean annual sunshine duration in 1979-2018 at Hornsund. (b) Variability of the monthly sums of sunshine duration at Hornsund in the period 1979-2018.**

Monthly total SD is presented in Figure 8b. Its variability results from the different duration of the day at the location (latitude
77N) with zero SD during the polar night.



### 3.7 Cloudiness

Arctic clouds have a warming effect on the surface during most of the year because their effect of increasing the downward longwave radiation dominates their effect of reducing the net solar radiation over high-albedo snow and ice surfaces. In summer, however, clouds typically have a cooling effect on surface types with a lower albedo, such as the open sea, melting sea ice, and ground (Intrieri et al., 2002; Shupe and Intrieri, 2004). Observations of cloudiness at the Polish Polar Station in Hornsund are conducted by meteorologists and describe the predominant sky condition based upon octas (eighths) of the sky covered by opaque (not transparent) clouds. There are many factors that may hinder the heterogeneity and evaluation of cloudiness, due to the annual change in the meteorological observers and a fact that observers might be subjective, although are provided with clear observable criteria.

Annual averages of cloudiness in the period 1983-2018 is presented in Figure 9a. The mean over this period equals 5.85 octas. The minimum value of annual was observed in 1988 (5.16 octas) and maximum in 1984 (6.39 octas). An increasing tendency of mean annual cloudiness is visible. The estimated trend (slope 0.13 octas/decade) is statistically significant at the 0.05 level.

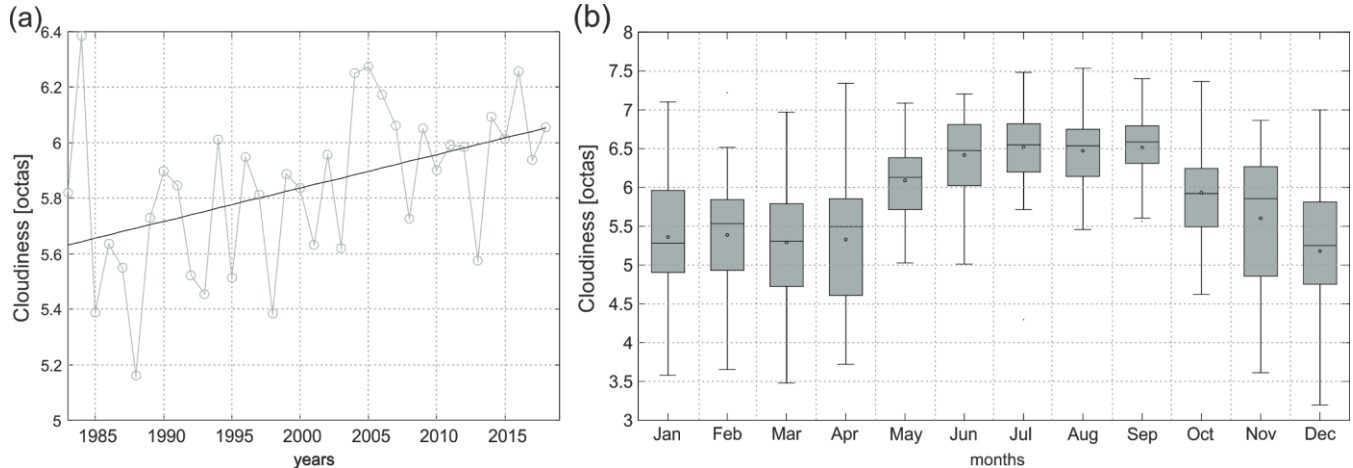

**Figure 9 (a) Variability of mean annual cloudiness in 1983-2018 at Hornsund. (b) Variability of the monthly sums of cloudiness at Hornsund in the period 1983-2018.**

The variability of the monthly cloudiness in the period 1983-2018 is presented in Figure 9b. The annual run is characterised by lower mean cloudiness during the cold period from October till April (5.5-6.0 octas), and it is also characterised by larger inter-annual variability. The period from May till September is on the average more cloudy (6.0-6.7 octas) and inter-annual variability is lower.

### 3.7 Visibility

The horizontal visibility is quantified using observations made by meteorologists in the surroundings of the Hornsund Station with a marine scale that range from 1 to 9. The visual observations are performed using known distances to the surrounding mountains and other objects. Values 1 and 2 correspond to very bad visibility, 0-50 m and 50-200 m, respectively. Bad



visibility (200 m – 1 km) is represented by value 3. Weak horizontal visibility represents conditions with 1-2 km and 2-4 km
that are quantified as 4 and 5 in the applied scale. Moderate horizontal visibility, described as 6 in the scale, represent conditions
when an object or light can be clearly discerned from 4-10 km. Good horizontal visibility (7 in the scale) is 10-20 km, very
good (8) 20-50 km and extremely good (9) is for horizontal visibility larger than 50 km. Noted visibility might be reduced by
multiple factors, including all products of the condensation of water vapour such as fog, precipitation, as well as darkness
during cloudy conditions throughout the polar night, as there are no artificial lights in the area. There are no anthropogenic
factors that would reduce visibility in the vicinity of the Hornsund Station as it is located in the middle of the strictly protected
South Spitsbergen National Park. Due to that reduced visibility cannot be an indicator of poor air quality.

Figure 10a shows the variability of mean annual visibility in the period 1983-2018. On average in this period is good horizontal
visibility that amounts 7.40; minimum mean annual visibility was observed in 2016 (7.08) while maximum in 1987 (7.70). A
decreasing tendency is visible (slope of trend -0.02 per decade) however the trend is not statistically significant at the 0.05
level. Variability of the mean monthly visibility at Hornsund in the period 1983-2018 is presented in Figure 10b. It is
characterised by both low inter-annual and interseasonal variability and on average reaches values between 7 and 8.

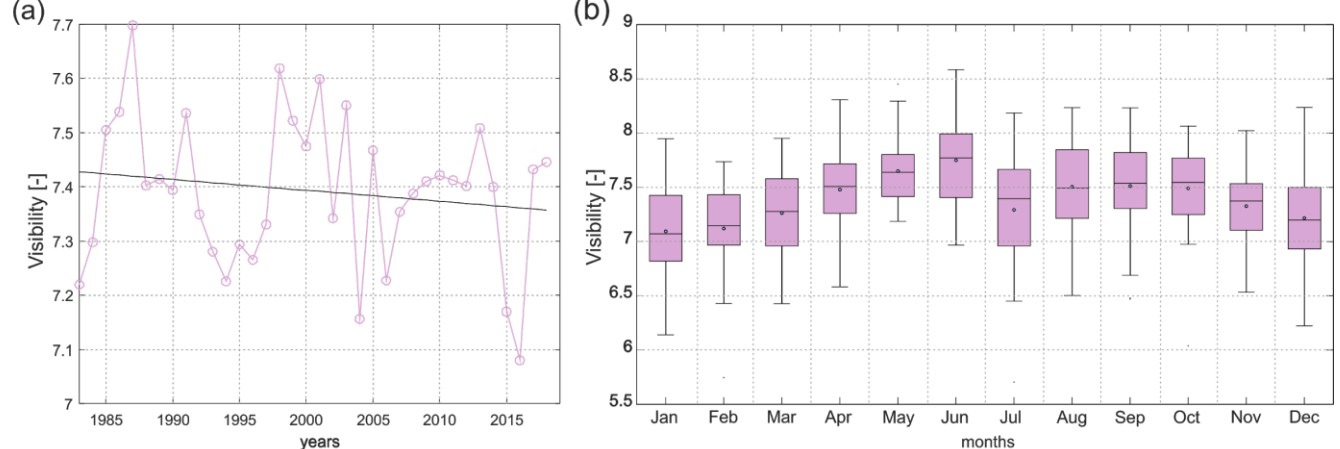

**Figure 10 (a) Variability of mean annual visibility in 1983-2018 at Hornsund. (b) Variability of the mean monthly visibility at Hornsund in the period 1983-2018.**

## 4 Quality control of the time series

All presented datasets have undergone a thorough quality control process. Such process consisted of multiple steps as the
measurements may not be homogenous due to the varying number of observations during the day, changes of sensors, and
other factors (Estévez et al. 2011). In the first step, the data were visualised as a time series that allowed verification if all data
have been collected and that the record structure is correct, complete, and without any gaps. In this way also the presence of
outliers and step change in the data was tested. In the following step, different variables were compared to test the internal

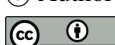



consistency between variables. Such analyses include a comparison of minimum, mean, and maximum daily TA that follow
the rule TAmax>TAmean>TAmin. In the case of WS and WD following conditions were tested WS=0 and WD=0, WS≠0 and
WD≠0. In the third step temporal consistency of time series was analysed with the help of statistical tests of homogeneity
(Pettit and Standard Normal Homogeneity Test). In the last step, the same variables but from different meteorological stations
in Svalbard were compared. The air temperature time series were tested against observations in Barentsburg, Bjørnøya, Hopen,
Longyearbyen (Svalbard Lufthavn), Ny Ålesund, and Sveagruva. For that purpose, the data were visualised and checked with
the Standard Normal Homogeneity Test (Alexandersson, 1986; Nordli et al., 1996). The applied algorithm showed good
performance in both detecting breakpoints and identifying homogeneous time series. By application of the relative method,
with comparison to the other available datasets from Svalbard, the gradual and step changes due to climate change were not
found as a source of inhomogeneity.

**5 Data availability**

The dataset described in this article is available on the PANGAEA repository (Wawrzyniak and Osuch, 2019:
https://doi.pangaea.de/10.1594/PANGAEA.909042).

**6 Summary**

This paper has presented details of a long-term (1979–2018) dataset from the meteorological site at the Polish Polar Station
Hornsund located in the SW part of Spitsbergen. The data series includes daily, monthly and annual air temperature, PDD,
NDD, the sum of precipitation, air humidity, atmospheric pressure, wind speed and direction, sunshine duration, cloudiness,
and visibility. This rich dataset, now available online, is a valuable source for documenting the state of the climate in SW
Spitsbergen that represents the Atlantic sector of the Arctic. Nowhere on the planet is climate warming faster than here. With
the positive trend of mean annual temperature $+1.14°$C/decade, the climate in Hornsund is warming five times faster than the
global average. All climatological variables presented in this study have many environmental implications and there is a broad
scientific interest and societal need to understand climate variability and its influence on geoecosystems.

**7 Author Contributions**

TW and MO wrote the paper and carried out the data processing and analysis.

**8 Competing Interests**

The authors declare that they have no conflict of interest.





**Acknowledgements**

The authors would like to kindly thank the meteorological staff from the Polish Polar Station Hornsund listed here: https://hornsund.igf.edu.pl/about-the-station/expeditions/ for collecting the data and maintaining the meteorological monitoring. Financial support for this work was provided by the Polish National Science Centre through grant No. 2017/27/B/ST10/01269. This work was also partially supported by the Institute of Geophysics, Polish Academy of Sciences within statutory activities No 3841/E-41/S/2019 of the Ministry of Science and Higher Education of Poland.

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
