# Peer review of "Forty years High Arctic climatological dataset of the Polish Polar"

_Earth System Science Data, 2019_

## Referee Comment (RC1) · Anonymous Referee #1 · 14 Jan 2020

In this paper, authors presented a unique meteorological dataset from an Arctic weather station in the Svalbard Archipelago for the period of 1979-2018. The paper is very well-organized and the dataset presented is a very useful contribution to the monitoring of remote northern regions that can be used for testing and evaluating hydrology, weather, and climate models and to validate remote sensing products. I recommend publishing the paper after a minor revision and addressing the following comments.

General comments:

1. I am wondering if the authors can provide information about snow depth and snow water equivalent, which are key elements for the Arctic environment. How about soil

temperature and moisture and permafrost information?

Specific comments:

1. Title: Replace "40" with "Forty"

2. Line 7: Replace "SW part" with "southwest".

3. Line 9-10: Start as a new sentence and rephrase it as "Therefore, the described time series of observations in this paper are of unique value."

4. Line 11 (also line 58): I would remove ", systematic," as we use "systematic" for errors and not observations. Also, add "and compare" after "analyse" in the line above and remove "and comparing the corresponding data". Change "rely" to "relying".

5. Line 16: Spell out "PANGAEA" or introduce it.

6. Line 19-20: Rephrase the sentence and remove the "if condition".

7. Line 23-24: Add references for this claims: "The climatic characteristics are primarily determined by astronomical factors, but there are differences in the mechanisms that cause a regional warming trend and determine their magnitude."

8. Line 26: Replace "ocean and atmospheric" with "oceanic and atmospheric" to be consistent.

9. Line 37: Change to read as "There are evidences for anomalies and changes in the recent years in western Spitsbergen, including higher are temperature ...".

10. Fig 1 caption: Add "in the Svalbard Archipelago" to the end of the caption

11. Line 46: Replace "became" with "has become".

12. Line 51: Use lower case for "Station" here and anywhere else (e.g., in line 53) that does not follow a name. Use short sentences with clear "subject" and "verb" in each sentence. Verbs like "covered, raised, and covered" need a subject. For instance, it can be used as "it is covered/raised".

13. Figure 2 and other figures if relevant: I am wondering if you can add the significant trend values on the figures.

14. Line 121: Check the grammar here and throughout the manuscript. Here, add "are" before "not considered".

15. Table 2: Are the numbers with a star sign statistically significant? The caption is confusing. You should assign the symbols for those that are statistically significant not the opposite. Define variable indices in the caption, e.g., What is VV? I suggest removing RH and maybe WS columns as they seem not statistically significant.

16. I suggest removing Figure 3 for relative humidity, because it is somehow presented later in Figure 5 in the form of atmospheric vapour pressure and it does not show any significant trends, not monthly nor annually.

17. Line 141 and throughout the paper: Replace "a lack of a statistically significant trend" with "a statistically insignificant trend".

18. Line 156: Rewrite as: "...creates a relatively moist climate in SW Spitsbergen region, which ...

19. Figure 7: Remove the trend line if it is insignificant (e.g., line 197). Same goes to Figure 8.

20. Figure 9 can be easily merged to Figures 8 and 10.

---

## Referee Comment (RC2) · Anonymous Referee #2 · 28 Jan 2020

The arcticle "40 years High Arctic climatological dataset of the Polish Polar Station Hornsund (SW Spitsbergen, Svalbard)" by Wawrzyniak and Osuch describes meteorological observation data from Hornsund Station on Svalbard. In the manuscript, the authors present both the climatological appearance of the meteorological variables and their change over time. The according data set is available with different temporal averaging resolution at the PANGAEA repository. The 40 year data record provides a valuable contribution to the monitoring of climate change in this Arctic region. The manuscript is well structured and needs only minor revision as listed below.

——————————————————————

[Figure]

— General Comments —

In addition to a climatological analysis, the authors present trends of various meteorological parameters. As trend analysis builds on homogeneous data series, the change of instrumentation during the observation period needs to be considered. For some parameters, the different instrumentation may have had different precision and uncertainties, while for others the time resolution of the measurements may have impacted the homogeneity (e.g. when switching from 3-hourly observations with a mercury barometer to 60-second observations with a Vaisala PTB200A sensor). While the general quality control of the time series is described, please add a statement on the data homogeneity of each individual parameter and its suitability to derive trends.
* * *
— Specific Comments —

- lines 22-27: climate change in the Arctic is not just reflecting the global warming trend, but is known to be amplified by various processes that go beyond astronomical factors. Please provide according references here.

- Lines 26-27: what about latent heat release from open water where there used to be sea ice cover ?

- Line 36-37: air temperature changes in Hornsund are the largest on Earth ? What about other stations on Svalbard ? Please put into context of Svalbard region (see e.g. Gjelten et al., Pol.Res.2016) and North Atlantic Arctic in general.

- Line 54-55: the maximum holocene extent of Hansbreen glacier does not add any useful information here; suggest to skip this sentence

- Line 57: 01003 seems to be the official WMO station number, so I suggest to provide the link to the OSCAR database (https://oscar.wmo.int/surface/#/) instead of ogimet

- Lines 94-95: does this citation of Hanssen-Bauer et al. (2019) really refer to the min-

imum, mean, and maximum TA increase found in the Hornsund station data ? Please set the context correctly.

- Line 130-131: when was the humidity sensor replaced ? This should also be mentioned in Table 1.

- Table 1: change in instrumentation needs to be documented more precisely, e.g. the change in pressure sensor is missing completely. Also the other instruments have likely not been changed at the turn of the year, so please provide at least the month of change. Annual mean values containing data from different instruments should be analyzed with care in regard of data homogeneity.

- Line 158: Given uncertainty issues with solid precipitation measurements (e.g. blowing snow, undercatch; see Forland & Hanssen-Bauer, Pol.Res.2013), please provide more details on the precipitation measurement set-up. Are there one or two fences installed around the Hellmann gauge ? Have any infrastructural changes occurred to the measurement site that may affect the blowing snow ?

- Table 2 : For some of the parameters, the given numbers pretend a precision that is not given by the measurements (e.g. relative humidity, precipitation). For the sunshine duration, the total possible duration for each month would be of interest, in particular for the month of February which is partly in the dark season but sees a considerable siginificant trend.

- Lines 171-172: Since 2001, the pressure measurements are taken every 60 seconds. Please describe how you retrieve the 3-hourly value (is it a mean of +/- 1.5 hours around the time step ?).

- Lines 205-206: The analysis of the atmospheric energy balance requires up- and downward components of the radiative flux, sensible heat flux, latent heat flux and momentum flux . . . Please correct your statement here.

- Lines 286-288: please add a reference

—————————————————————————

— Technical Corrections —

- Line 7: Southwest instead of SW

- Line 150: The minimum observed quantity . . .

- Line 232: The minimum value of annual mean was observed. . .

- Line 237: The annual cycle. . .

- Line 243: . . . with a marine scale that ranges. . .

- Line 252: add ". . . on the local scale." (consider possible advection of pollution)

---

## Author Comment (AC1) · 28 Feb 2020

We appreciate the reviewer comments as they provide valuable feedback to increase the quality of this paper. We revised our manuscript and corrected the text and figures and included additional descriptions according to the suggestions. Detailed answers are included below. Referee text (R) and author responses (A) are indicated.

R: General comments: 1. I am wondering if the authors can provide information about snow depth and snow water equivalent, which are key elements for the Arctic environment. How about soil temperature and moisture and permafrost information?

[Figure]

A: The time series of snow depth was presented in Osuch, M., & Wawrzyniak, T. (2017). Variations and changes in snow depth at meteorological stations Barentsburg and Hornsund (Spitsbergen). Annals of Glaciology, 58(75pt1), 11-20. doi:10.1017/aog.2017.20 So far the work on historical ground temperature data (from the borehole 1 m deep) is in progress and as new (10, 12 and 20 m deep) boreholes were established two years ago in Hornsund we are preparing an article on modelling of the ground thermal regime in deeper layers. There are plans to measure soil moisture but so far there is none.

Specific comments:

R: 1. Title: Replace "40" with "Forty"

A: Corrected

R: 2. Line 7: Replace "SW part" with "southwest".

A: Corrected.

R: 3. Line 9-10: Start as a new sentence and rephrase it as "Therefore, the described time series of observations in this paper are of unique value."

A: Corrected.

R: 4. Line 11 (also line 58): I would remove ", systematic," as we use "systematic" for errors and not observations. Also, add "and compare" after "analyse" in the line above and remove "and comparing the corresponding data". Change "rely" to "relying".

A: Corrected.

R: 5. Line 16: Spell out "PANGAEA" or introduce it.

A: PANGAEA is a Data Publisher for Earth & Environmental Science, that in our opinion is a recognizable repository of scientific data.

R: 6. Line 19-20: Rephrase the sentence and remove the "if condition".

A: Rewritten "For the analysis of the Arctic climate change, the long term operational monitoring of meteorological variables including reliable observations and measurements is obligatory."

R: 7. Line 23-24: Add references for this claims: "The climatic characteristics are primarily determined by astronomical factors, but there are differences in the mechanisms that cause a regional warming trend and determine their magnitude."

A: We corrected the text and added missing reference: Mechanisms of Arctic amplification are still not fully understood but include feedback of reduced summer albedo due to reduction of sea ice extent and snow cover loss, higher sea surface temperatures, an increase of atmospheric water vapour content, cloud conditions, and changes in atmospheric circulation (IPCC, 2019).

R: 8. Line 26: Replace "ocean and atmospheric" with "oceanic and atmospheric" to be consistent.

A: Corrected.

R: 9. Line 37: Change to read as "There are evidences for anomalies and changes in the recent years in western Spitsbergen, including higher are temperature ...".

A: Corrected.

R: 10. Fig 1 caption: Add "in the Svalbard Archipelago" to the end of the caption.

A: Corrected.

R: 11. Line 46: Replace "became" with "has become".

A: Corrected.

R: 12. Line 51: Use lower case for "Station" here and anywhere else (e.g., in line 53) that does not follow a name. Use short sentences with clear "subject" and "verb" in each sentence. Verbs like "covered, raised, and covered" need a subject. For instance,

it can be used as "it is covered/raised".

A: Corrected.

R: 13. Figure 2 and other figures if relevant: I am wondering if you can add the significant trend values on the figures.

A: Added.

R: 14. Line 121: Check the grammar here and throughout the manuscript. Here, add "are" before "not considered".

A: Corrected.

R: 15. Table 2: Are the numbers with a star sign statistically significant? The caption is confusing. You should assign the symbols for those that are statistically significant not the opposite. Define variable indices in the caption, e.g., What is VV?

A: Changed, we added the description of abbreviations.

R: I suggest removing RH and maybe WS columns as they seem not statistically significant.

A: We would like to show the trends of every analyzed variable in this table, as in our opinion the table serves as a summary of the estimated trends.

R: 16. I suggest removing Figure 3 for relative humidity, because it is somehow presented later in Figure 5 in the form of atmospheric vapour pressure and it does not show any significant trends, not monthly nor annually.

A: Thank you for this comment. The structure of this article is organized with variables and relative humidity and atmospheric pressure and we really want to keep all variables in the presented order.

R: 17. Line 141 and throughout the paper: Replace "a lack of a statistically significant trend" with "a statistically insignificant trend".

A: Corrected.

R: 18. Line 156: Rewrite as: "...creates a relatively moist climate in SW Spitsbergen region, which ...

A: Corrected.

R: 19. Figure 7: Remove the trend line if it is insignificant (e.g., line 197). Same goes to Figure 8.

A: Changed.

R: 20. Figure 9 can be easily merged to Figures 8 and 10.

A: We would like to keep the order and the structure of the paper – to present each variable in the same manner.

---

## Author Comment (AC2) · 28 Feb 2020

We appreciate the reviewer comments as they provide valuable feedback to increase the quality of this paper. We revised our manuscript and corrected the text and figures and included additional descriptions according to the suggestions. Detailed answers are included below. Referee text (R) and author responses (A) are indicated.

General Comments

R: In addition to a climatological analysis, the authors present trends of various meteorological parameters. As trend analysis builds on homogeneous data series, the

change of instrumentation during the observation period needs to be considered. For some parameters, the different instrumentation may have had different precision and uncertainties, while for others the time resolution of the measurements may have impacted the homogeneity (e.g. when switching from 3-hourly observations with a mercury barometer to 60-second observations with a Vaisala PTB200A sensor). While the general quality control of the time series is described, please add a statement on the data homogeneity of each individual parameter and its suitability to derive trends.

A: We described the changes in instrumentation and added the information that the old and new sensors were operating simultaneously for more than one year to compare the results and determine the degree of compatibility and homogeneity of the measurements.

Specific Comments

R: lines 22-27: climate change in the Arctic is not just reflecting the global warming trend, but is known to be amplified by various processes that go beyond astronomical factors. Please provide according references here Lines 26-27: what about latent heat release from open water where there used to be sea ice cover?

A: We extended the text: Mechanisms of Arctic amplification are still not fully understood but include feedback of reduced summer albedo due to reduction of sea ice extent and snow cover loss, higher sea surface temperatures, an increase of atmospheric water vapor content, cloud conditions, and changes in atmospheric circulation (IPCC, 2019).

R: Line 36-37: air temperature changes in Hornsund are the largest on Earth ? What about other stations on Svalbard ? Please put into context of Svalbard region (see e.g. Gjelten et al., Pol.Res.2016) and North Atlantic Arctic in general.

A: We corrected the text: Relatively to the other parts of the Arctic, air temperatures in Svalbard are the highest at this latitude and their observed changes are one of the

largest on Earth. There is evidence for anomalies and changes in recent years in Atlantic sector of the Arctic along western Spitsbergen, including higher air temperature (Gjelten et al. 2016) and higher liquid precipitation (Osuch and Wawrzyniak, 2017a). We compared the results of trend analysis for different meteorological stations in the period 1979-2018, including those located in Svalbard (Hornsund, Barentsburg, Bjornoya, Hopen, Lufthavn, NyAlesund, Sveagruva) and you are right. The results are as listed below: – trends [°C/decade] Hornsund (1.14); Barentsburg (0.97); Bjornoya (0.71); Hopen (1.15); Lufthavn (1.13); NyAlesund (0.96); Sveagruva (1.06).

Gjelten et al. in Pol.Res. 2016 presented trends for the period 1979-2015 – different than in our study (1979-2018), so the results are slightly different.

R: Line 54-55: the maximum holocene extent of Hansbreen glacier does not add any useful information here; suggest to skip this sentence.

A: Removed.

R: Line 57: 01003 seems to be the official WMO station number, so I suggest to provide the link to the OSCAR database (https://oscar.wmo.int/surface/#/) instead of ogimet.

A: Changed.

R: Lines 94-95: does this citation of Hanssen-Bauer et al. (2019) really refer to the minimum, mean, and maximum TA increase found in the Hornsund station data ? Please set the context correctly.

A: We corrected the text: The estimated slope of trend equal to 1.34, 1.14, and 1.00°C/decade for minimum, mean, and maximum TA respectively. These are one of the highest increases of mean TA on the planet, more than six times larger than the global average of +0.17°C per decade (NOAA, 2020). The rest of the world is not expected to experience such changes until the end of this century (Hanssen-Bauer et al., 2019).

R: Line 130-131: when was the humidity sensor replaced ? This should also be mentioned in Table 1. - Table 1: change in instrumentation needs to be documented more precisely, e.g. the change in pressure sensor is missing completely. Also the other instruments have likely not been changed at the turn of the year, so please provide at least the month of change. Annual mean values containing data from different instruments should be analyzed with care in regard of data homogeneity.

A: We added the information on the changes of instruments in Table 1 and added to the text: Since January 2001 most of the traditional instruments were replaced by an automatic weather station with Vaisala QLC-50 logger. The sensors of the new system have been installed on meteorological mast, situated 160 m SW of the main station building. To replace Vaisala QLC-50 in September 2016 new system Vaisala MAWS 301 was set on the same meteorological mast. To determine the degree of compatibility and homogeneity of the measurements, the old and new sensors were operating simultaneously for more than one year. The results of the analysis allowed to combine time series and since January 2018 the data comes from Vaisala MAWS 301. A comprehensive description of measurements and instruments can be found in collective work edited by Marsz and Styszyńska (2013) and in Table 1.

R: Line 158: Given uncertainty issues with solid precipitation measurements (e.g. blowing snow, undercatch; see Forland & Hanssen-Bauer, Pol.Res.2013), please provide more details on the precipitation measurement set-up. Are there one or two fences installed around the Hellmann gauge?

A: We added the information that Hellman gauge at Hornsund station is unfenced. We are recently working on the article that compares data from three different precipitation gauges installed in Hornsund in last years (one fence Geonor and Parsivel), though for long period only the Hellman gauge has been used in the same, unchanged spot. We present raw data of measured precipitation, that can be recalculated to take into account eg wind-induced undercatch with wind speed data that is also provided.

R: Have any infrastructural changes occurred to the measurement site that may affect
the blowing snow?

A: Redevelopment of the station's main building in 2004, when a northern wing was added, is a potential factor that could influence measurements. However, we did not find significant step change in snow depth nor in precipitation amount after 2004, suggesting that this event had a minimal effect on the measurements. The rain gauge is placed some 60 m from the station's buildings. We analyzed the impact of changes in infrastructure on the snow depth here: - Osuch, M., & Wawrzyniak, T. (2017). Variations and changes in snow depth at meteorological stations Barentsburg and Hornsund (Spitsbergen). Annals of Glaciology, 58(75pt1), 11-20. doi:10.1017/aog.2017.20

R: Table 2 : For some of the parameters, the given numbers pretend a precision that is not given by the measurements (e.g. relative humidity, precipitation).

A: It is not the precision of the instruments but the slope of trend rounded to two decimal places.

R: For the sunshine duration, the total possible duration for each month would be of interest, in particular for the month of February which is partly in the dark season but sees a considerable siginificant trend.

A: That is true and it was studied by Wojkowski et al. 2015, that may be found in the reference list.

R: Lines 171-172: Since 2001, the pressure measurements are taken every 60 seconds. Please describe how you retrieve the 3-hourly value (is it a mean of +/- 1.5 hours around the time step?).

A: These are the instantaneous data from given time.

R: Lines 205-206: The analysis of the atmospheric energy balance requires up- and downward components of the radiative flux, sensible heat flux, latent heat flux and momentum flux . . . Please correct your statement here.

A: That is true, we changed the text: Sunshine duration (SD) is one of the important meteorological variables that provides data on the time period during which direct solar radiation reaches the Earth's surface and partly on the quantity of total solar energy.

R: Lines 286-288: please add a reference.

A: We added the reference: NOAA National Centers for Environmental information, Climate at a Glance: Global Time Series, published February 2020, retrieved on February 27, 2020 from https://www.ncdc.noaa.gov/cag/, 2020.

Technical Corrections

R: Line 7: Southwest instead of SW.

A: Corrected.

R: Line 150: The minimum observed quantity . .

A: Corrected.

R: Line 232: The minimum value of annual mean was observed. . .

A: Corrected.

R: Line 237: The annual cycle. . .

A: Corrected.

R: Line 243: . . . with a marine scale that ranges. . .

A: Corrected.

R: Line 252: add ". . . on the local scale." (consider possible advection of pollution).

A: Corrected.
* * *